# The Oncoprotein Fra-2 Drives the Activation of Human Endogenous Retrovirus Env Expression in Adult T-Cell Leukemia/Lymphoma (ATLL) Patients

**DOI:** 10.3390/cells13181517

**Published:** 2024-09-10

**Authors:** Julie Tram, Laetitia Marty, Célima Mourouvin, Magali Abrantes, Ilham Jaafari, Raymond Césaire, Philippe Hélias, Benoit Barbeau, Jean-Michel Mesnard, Véronique Baccini, Laurent Chaloin, Jean-Marie Jr. Peloponese

**Affiliations:** 1Université Montpellier (UM), 34000 Montpellier, France; jtram@bwh.harvard.edu (J.T.); laetitia.marty@irim.cnrs.fr (L.M.); mourouvin.celima@gmail.com (C.M.); magali.abrantes@irim.cnrs.fr (M.A.); laurent.chaloin@irim.cnrs.fr (L.C.); 2Institut de Recherche en Infectiologie de Montpellier (IRIM), CNRS, 34293 Montpellier, France; 3Centre Hospitalier Universitaire de Martinique, 97261 Fort de France, France; 4Département de Radiothérapie-Oncologie-Hématologie, Centre Hospitalier Universitaire de la Guadeloupe, 97110 Pointe à Pitre, France; philippe.helias@chu-guadeloupe.fr; 5Département des Sciences Biologiques, Université du Québec à Montréal, SB-R860, Montréal, QC H2X 1Y4, Canada; barbeau.benoit@uqam.ca; 6Laboratoire d’Hématologie CHU de la Guadeloupe, 97110 Pointe à Pitre Guadeloupe, France; veronique.baccini@chu-guadeloupe.fr

**Keywords:** HERVs, leukemia, ATLL, AP-1, Fra-2

## Abstract

Human endogenous retroviruses (HERVs) are retroviral sequences integrated into 8% of the human genome resulting from ancient exogenous retroviral infections. Unlike endogenous retroviruses of other mammalian species, HERVs are mostly replication and retro-transposition defective, and their transcription is strictly regulated by epigenetic mechanisms in normal cells. A significant addition to the growing body of research reveals that HERVs’ aberrant activation is often associated with offsetting diseases like autoimmunity, neurodegenerative diseases, cancers, and chemoresistance. Adult T-cell leukemia/lymphoma (ATLL) is a very aggressive and chemoresistant leukemia caused by the human T-cell leukemia virus type 1 (HTLV-1). The prognosis of ATLL remains poor despite several new agents being approved in the last few years. In the present study, we compare the expression of HERV genes in CD8^+^-depleted PBMCs from HTLV-1 asymptomatic carriers and patients with acute ATLL. Herein, we show that HERVs are highly upregulated in acute ATLL. Our results further demonstrate that the oncoprotein Fra-2 binds the LTR region and activates the transcription of several HERV families, including HERV-H and HERV-K families. This raises the exciting possibility that upregulated HERV expression could be a key factor in ATLL development and the observed chemoresistance, potentially leading to new therapeutic strategies and significantly impacting the field of oncology and virology.

## 1. Introduction

Adult T-cell leukemia/lymphoma (ATLL) is caused by human T-cell leukemia virus-1 (HTLV-1), which is also the etiologic agent of HTLV-1-associated myelopathy/tropical spastic paraparesis (HAM/TSP). The estimated lifetime risk of developing ATLL in HTLV-1 carriers is 2–7%, and the disease occurs at least 20–30 years after HTLV-1 infection [1,2,3]. ATLL is classified as a peripheral T-lymphocytic malignancy with a CD4^+^ T phenotype. The diversity in clinical features and evolution has led to the classification into four clinical subtypes: smoldering, chronic, acute, and lymphoma-type ATLL [4]. Patients with acute or lymphoma forms have high-risk ATLL (HR-ATLL), which is associated with a poor prognosis due to multidrug resistance of ATLL cells, rapid proliferation, significant tumor burden, hypercalcemia, and infectious complications linked to reduced immunologic competence [2,5,6,7]. Despite the introduction of antiviral therapy, such as zidovudine and interferon alpha in combination with chemotherapy, the prognosis of HR-ATLL remains very poor, with a three-year survival rate of less than 30% and a high relapse rate [1,8]. New diagnostic and therapeutic approaches are direly needed.

Human endogenous retroviruses (HERVs), which arise from the infection of human germ-line cells by exogenous retrovirus during the evolution of primates, account for approximately 8% of the human genome [9,10,11]. HERVs are classified into three classes based on sequence similarities to different infectious retroviruses: gamma retrovirus (class I), beta retrovirus (class II), and spumaretrovirus like (class III) [9,10,11]. Each class is also divided into subgroups based on the specificity of the tRNA primer-binding site (PBS) used during retroviral replication [9,10,11]. While the majority of HERVs lack infectious capacity due to the accumulation of deletions, mutations, and insertions in internal coding regions and long terminal repeats (LTRs), recent studies have shown that HERV LTRs are still contributing to the human transcriptome [9,10,11]. Increasing evidence suggests that the expression of HERV proteins, including the envelope (Env), contributes to complex pathologies, such as autoimmune diseases and cancer [9,10,11]. However, the role of HERV Env in those pathologies is still controversial. It has been shown that HERV Env can either trigger innate and adaptive immunity, thus promoting inflammatory and cytotoxic reactions, or act as immune downregulators by preventing immune response activation [9,10,11]. Furthermore, HERV Env has been proposed to contribute to tumor development, metastases, and intrinsic chemoresistance of cancer cells by inducing abnormal cell–cell fusion [12].

Here, we report that the expression of several HERVs is upregulated in ATLL cell lines and PBMCs from acute ATLL patients. We also observed that while HBZ has little effect, the oncoprotein Fra-2 (or FosL2), but not cFos, can transactivate the LTR of HERV-H-Hu13, HERV-E, HERV-L, and HERV-K (HML-4). Furthermore, the co-expression of HBZ does not impair the activation of the different HERV-LTRs. Using a chromatin immunoprecipitation assay, we further demonstrate that Fra-2 binds on the LTR region of several HERVs-H. We measured the antibody response against different HERV Env in ATLL patient sera. We observed a more robust antibody response against HERV Env compared to sera of asymptomatic carriers. Further studies are needed to determine whether anti-HERV antibodies could be a promising early biomarker for predicting eventual disease development.

## 2. Materials and Methods

### 2.1. HTLV-1 Patients

The HTLV-1 infected patients’ samples, including eight non-infected individuals, twelve HTLV-1-infected patients without declared malignancy (AC), and telve acute ATLL patients (ATLL), were collected at the University Hospital of Fort-de-France, Martinique, and the University Hospital of Pointe à Pitre, Guadeloupe (French West Indies) (NCT01754311, NCT01941680). Serum and blood samples were obtained from the processing of biological samples by the Centre de Ressources Biologiques of Martinique (CeRBiM) (BRIF BB-0033–00099) (CHU Martinique) and the Centre de Ressources Biologiques of Guadeloupe (KaruBioTech) (DC-2016–2827) (CHU of Guadeloupe). The diagnosis of ATLL was based on clinical features, hematological findings, the presence of the HTLV-1 provirus in leukemic cells, and the detection of anti-HTLV-1 antibodies in serum. ATLL was subtyped according to the JLSG criteria. Peripheral blood samples were used following French bioethics laws concerning biologic samples.

### 2.2. Materials

All HERV-LTR plasmids are from Schön et al. 2001 [13]. pcDNA 3,1-HA-Fra-2 (OHu30302) was purchased from GenScript Biotech (Rijswijk, The Netherlands). pCDNA5-FRT-3Flag-JunB (human), pCDNA5-FRT-3Flag-cJun (human), and pCDNA5-FRT-HA-cFos (human) were provided by the MGC platform (Biocampus, Montpellier, France). pcAG-Tax-Flag, pCMV-P65-Flag, pcDNA3.1 HBZ-Myc, and pcDNA 3.1 JunD-Flag were previously described [14,15].

### 2.3. Cell Culture and Transfection

Human T-cell lines Jurkat (ATCC^®^ TIB-152™), HUT-102 (ATCC^®^ TIB-162™), C8166 (ECACC 88051601), and ATL-2 (-) (CVCL_A6TF) were propagated in RPMI-1640 with 10% fetal calf serum (FCS). HEK 293T cells (ATCC^®^ CRL-3216™) were propagated in DMEM with 10% FCS and transfected according to the manufacturer’s protocol using PEI MAX (1 mg/mL). To stably express Fra-2 in HEK 293, we transfected them with three µg of pcDNA 3.1-HA-Fra-2 using PEI MAX and selected them with G418 (500 µg/mL) for three weeks. Clones were then pooled. CD8^+^-deprived PBMCs of HTLV-1-infected patients were isolated and cultured, as previously described [16].

### 2.4. RNA Analysis

Cells were collected and cryopreserved as dry pellets until used. Nucleic acid was extracted using the Qiagen AllPrep DNA/RNA Mini Kit (Qiagen, Courtaboeuf, France). To obtain first-strand cDNA, total RNA isolated from each sample was subjected to reverse transcription using the All-In-One 5X RT MasterMix (abm) according to the manufacturer’s protocol. The abundance of HERV transcripts was assessed by real-time quantitative (q) PCR analysis using SYBR Green I Master Mix and gene-specific primer sets (Appendix A). Standard curves were generated from each PCR plate for all primer pairs using serial dilution of an appropriate experimental sample. Samples were amplified in triplicate on each plate. Data were analyzed using LightCycler^®^480 Software (sw 1.5.1) (Roche Diagnostics, Meylan, France). Relative HERV mRNA levels among experimental samples were determined by the 2-ΔΔCT method, with values normalized to HPRT1 and GADPH as the reference housekeeping genes [17,18].

### 2.5. TF Binding Site Prediction

The different transcription factor binding sites in the HERV-LTR regions of HERV-W6 (AF315116.1), HERV-K (AB052568.1), HERV-H-MC16 (AF315098.1), HERV-H-L19 (AF315094.1), HERV-H-MP20 (AF315100.1), HERV-H-CL2 (AF315088.1), HERV-H-HU8 (AF315093.1), HERV-H-HU13 (AF315092.1), and HERV-L (X89211.1) were analyzed using Tfsitescan (Institute for Transcriptional Informatics, Pittsburgh, PA, USA) (http://www.ifti.org/cgi-bin/ifti/Tfsitescan.pl, accessed on 1 July 2024) and PROMO [19] (https://alggen.lsi.upc.es/cgi-bin/promo_v3/promo/promoinit.cgi?dirDB=TF_8.3, accessed on 1 July 2024).

### 2.6. Luciferase Assay

Luciferase assays were performed as previously described [20] in a Tecan Spark 10M microplate luminometer (Tecan, Lyon, France) with the Genofax A and Genofax C kit (Yelen, Ensues la Redonne, France) according to the manufacturer’s instructions. Firefly luciferase activities were normalized for transfection efficiency based on Renilla luciferase activity.

### 2.7. Western Blotting

Whole-cell lysates were prepared using a RIPA buffer [10 mM Tris–HCl (pH 7.4), 150 mM NaCl, 1% NP-40, one mM EDTA, 0.1% SDS, and one mM DTT] separated by electrophoresis on 12% SDS-PAGE gels (Bio Rad, Marnes la Coquette, France) and transferred onto PVDF membranes (Bio Rad, France). Incubations with primary antibodies to detect Fra-2 L-15 (sc-171) (Santa Cruz Biotech, Heidelberg, Germany), Flag M2 (F3165), Myc9E10 (M5546), HA (H3663), or β-actin (A3853) (Sigma-Aldrich, Saint-Quentin Fallavier, France) were followed by incubations with the appropriate secondary antibody conjugated with horseradish peroxidase (HRP) (Jackson Immunoresearch Europe Ltd., Ely, UK) and by detection with the enhanced luminescence (Roche Diagnostics France, Meylan, France) on a Chemidoc Imaging Systems (version 2.3.0.07) (Bio Rad, France).

### 2.8. ChIP-qPCR

According to the manufacturer’s instructions, ChIP experiments were performed with the Magna ChIP™ A/G Chromatin Immunoprecipitation Kit (Millipore, st quentin en yveline). Approximately 107 ATL-2 cells were harvested for ChIP. Briefly, cells were cross-linked with 1% formaldehyde for 10 min at room temperature. Glycine was added to stop the cross-linking reaction, and cells were washed with ice-cold PBS. Following solution removal, the tubes were chilled on ice, and cells were lysed using an ice-cold cell lysis buffer containing protease inhibitors. The chromatin was fragmented to 200 to 500 bp with the Micrococcal Nuclease Solution (Thermo Fisher Scientific Biosciences Gmbh, Villebon sur Yvette, France) at 37 °C for 20 min. After centrifugation, the supernatants were diluted with a cold dilution buffer containing protease inhibitors. Subsequently, five µg of the Fra-2 antibody (Fra2 (G-5X)-SC166102X) and control IgG were incubated at 4 °C overnight with the chromatin supernatants and protein A/G magnetic beads. After incubation, the antibody–chromatin/bead complex was washed four times, and DNA was purified with DNA-purifying Spin Filters. Immunoprecipitated DNA was used for qPCR analysis using the SYBR green detection method. ChIP-qPCR primers are shown in Appendix A.

### 2.9. ELISA Assay

Sera derived from non-infected patients (NI), asymptomatic carriers (ACs), and ATLL patients were collected before anti-cancer treatment using a standardized protocol and stored at −80 °C until use. The presence of antibodies against HERV Env proteins was determined by ELISA using the Qualitative Human HERVs Envelop Antibody (Anti-HERVS) ELISA Kit (from Mybiosource LLC, San Diego, CA, USA) according to the manufacturer’s instructions. Briefly, 50 μL of undiluted sera were added to each well and incubated for 90 min at 37 °C. Samples were washed four times with the provided wash buffer. Equal volumes of Chromogen Solution A and Chromogen Solution B were added to each well and incubated for 15 min at 37 °C. OD measurement at a wavelength of 450 nm was performed using a Tecan Spark 10M microplate reader within 5 min after adding the Stop Solution.

### 2.10. Statistical Analyses

Data on paired and unpaired observations were compared with a one-way ANOVA test with Dunn‘s multiple comparisons post-test and the Mann–Whitney *t*-test using GraphPad PRISM 8 software (version 8.4.3). Error bars indicate SEM. Differences were considered at ns *p* > 0.05 ** *p* < 0.01, *** *p* < 0.001 and **** *p* < 0.0001.

## 3. Results

### 3.1. HERV Env Proteins Induce a Humoral Response in Acute ATLL Patients

An increase in HERV gene expression could represent a potential diagnostic or prognostic cancer biomarker [21,22,23]. For this reason, using an enzyme-linked immunosorbent assay, we first examined the humoral response against different HERVs Env in the sera of patients with acute adult T-cell leukemia/lymphoma (ATLL) compared to HTLV-1 asymptomatic carriers (ACs) and a non-infected control (NI). We observed that ATLL patients’ sera exhibit elevated IgG levels reactive to HERV Env over the control sera of non-infected patients (NI) and HTLV-1 asymptomatic carriers (ACs) (Figure 1A). Next, we used receiver operating characteristic (ROC) curves to assess the accuracy of our ELISA assay and distinguish the ATLL groups from the AC control group. AC sera were compared to ATLL (Figure 1B). For ATLL, the area under the ROC curve (AUC) was 0.9796, and the 95% confidence interval was 0.9170 to 1.000 with a *p*-value of 0.0027. Our observation suggests that the titration of anti-HERV antibodies in the sera of HTLV-1 carriers could be a candidate biomarker for developing the disease. Unfortunately, the ELISA assay did not allow us to discriminate the subtype of expressed HERVs Env in ATLL patients.

### 3.2. HERV Gene Expression Levels Are Increased in ATLL Patients

A higher expression of human endogenous retroviruses (HERVs) has been associated with several malignancies, such as cancer [9,11,24]. However, little is known about the expression of HERVs in ATLL. We next evaluated the expression pattern of HERV genes, which have been described to be implicated in various human cancers, such as HERV-H, HERV-K (HML-6), HERV-R, and HERV-E [10,12,25,26,27]. These analyses were conducted on a control T-cell leukemic cell line (Jurkat), two HTLV-1-derived T-cell lines (HuT102, C81-66), and one cell line derived from an acute ATLL patient (ATL-2) (Figure 2). This analysis revealed an overexpression pattern of most HERV transcripts analyzed in ATL-2 cells compared to Jurkat or HuT102 (Figure 2).

Next, we compared the expression levels of HERV genes in cultured primary CD8^+^ deprived PBMCs freshly isolated from twelve asymptomatic carriers (ACs) vs. twelve ATLL patients (Figure 3). Similar to our observations in HTLV-1-infected cell lines, HERV-H Env, HERV-K (HML-6) Env, HERV-R Env, HERV-E gag HERV-K gag, and HERV-R pol transcript levels were statistically higher in ATLL patient cells compared to ACs (*p* = 0.0001) (Figure 3A–C,E–G). At the same time, no difference was observed for the HERV-W Env transcript (Figure 3D). Next, we performed a correlation analysis to assess a potential correlation between HERV mRNA levels and proviral load (PVL) in HTLV-1-infected individuals (Appendix A). No significant correlation was highlighted between the HTLV-1 PVL and the upregulation of HERV transcripts.

### 3.3. The Oncoprotein Fra-2 Activates the LTR of HERV-H, -E, -L, and -K

ERV expression can be upregulated by environmental factors, such as chemical substances, therapeutic agents, and viral infections [28,29,30,31]. Epigenetic mechanisms, including DNA methylation and histone modifications, as well as regulatory proteins, such as host transcription factors, cytokines, and small RNAs, can also be involved in upregulating HERV expression [28,29,30,31]. We first hypothesized that HTLV-1-encoded Tax and HBZ could regulate HERV expression in ATLL cells. Tax is a potent transcriptional activator of the viral LTR, which depends on recruiting CREB/ATF factors to the 5′LTR TRE1 elements [32,33,34]. HBZ has also been shown to regulate transcription by binding to the HTLV-1 3′LTR in complex with JunD [20]. To probe for the implication of Tax or HBZ in HERV transcription, we used a luciferase reporter assay to compare the ability of Tax and HBZ to activate transcription of HERV-H (Hu13), HERV-E, HERV-L, and HERV-K (HML-6) [13] in HEK293T cells. While both p65 (NF-kB subunit) and Tax significantly activated transcription from the HERV-H promoter in HEK293T (Figure 4C), HBZ did not increase the transcription from any of the four tested HERV promoters (Figure 4). Interestingly, Tax expression was absent in the tested ATLL cells, showing a critical level of HERV transcription (as shown in Figure 2H and Figure 3H). These observations suggest that neither HTLV-1 Tax nor HBZ is the primary activator of HERV transcription in ATLL cells.

To elucidate the mechanism by which transcription of the different HERVs was upregulated in ATLL cells, we examined the potential binding of transcription factors throughout the different HERV-LTR regions using Tfsitescan and PROMO [19]. We identified AP-1-binding sites within the U3-R region of HERV-H LTRs. We transfected HEK293T cells with constructs containing different HERV-LTR-driven luciferase genes and a panel of AP-1 expression vectors containing either c-Fos dimers or Fra-2 dimers (Figure 5). First, we confirmed that the transcription of the collagenase (MMP-1 gene) promoter was activated by all the cFos- and Fra-2-containing dimers (Figure 5A). We also observed that in transfected cells, Fra-2 dimers strongly activate the transcription of all four HERV-LTRs-Luc, while the c-Fos-containing dimers did not (Figure 5B–E). The structure of HERV-H LTRs varies and can be grouped into three subfamilies (type I, Ia, and type II) [13]. While all type I and Ia LTRs contain a GC-rich region downstream of the TATA box, the type I LTRs comprise a type I repeat region and a unique region I. In contrast, type Ia LTRs contain one type I repeat, several type II repeats, and a particular region I. In comparison, type II HERV-H LTRs contain three or four type II repeats and a unique region II [13] (Appendix A). Next, we assessed whether Fra-2 dimers activate transcription from several HERV-H members. In HEK293T, the co-expression of Fra-2 with Jun family members strongly increased luciferase activity from all tested HERV-H LTRs (Appendix A).

Since HBZ also binds to Jun family members, we assessed whether HBZ might impair Fra-2-mediated trans-activation of the different HERV-H promoters (Figure 6). The co-expression of HBZ with Fra-2 and other Jun members did not significantly decrease luciferase activity compared to Fra-2-only-expressing cells (Figure 6C–E).

### 3.4. Fra-2 Upregulates HERV-H Env and HERV-K Env Genes in Primary Cells Isolated from ATLL Patients

To evaluate the physiological relevance of our study, we first assessed the mRNA levels of HERV-H Env, HERV-R Env, HERV-K Env, and HERV-E gag in HEK293T cells stably expressing Fra-2 (Figure 7A). We found that the presence of Fra-2 was associated with a significant increase in HERV-H Env (*p* = 0.0087) and HERV-R Env (*p* = 0.0411) mRNA levels (Figure 7B,C).

Next, we cultivated CD8^+^-depleted PBMCs isolated from five patients with acute ATLL for five days and monitored transcript levels of Fra-2, HERV-H Env, and HERV-K Env, as well as Tax and HBZ (Figure 8, Appendix A) by qRT-PCR. As previously reported [35], HBZ mRNA increased over time, while Tax expression was very low, near the detection limit. Interestingly, during this timeframe, Fra-2 mRNA levels also significantly increased (*p* < 0.0001) (Figure 8A). Increases in HERV-H Env and HERV-K Env mRNA levels correlated with those of Fra-2 mRNA (HERV-H Env: R^2^ = 0.94; *p* < 0.001; HERV-K Env: R^2^ = 0.90; *p* < 0.001) (Figure 8B–E). These results suggest that in ATLL cells, the expression of AP-1 factors could lead to the upregulation of HERV-H Env and HERV-K Env transcription.

### 3.5. In Vivo Recruitment of Fra-2 to the HERV-H LTR

We next assessed the recruitment of Fra-2 in vivo within the physiologically relevant context of the chromatin structure by performing chromatin immunoprecipitation (ChIP) experiments on nuclear extracts of ATL-2 since this cell line expresses a high level of both Fra-2 and HERV-H Env mRNA (Figure 1). ChIP with anti-Fra-2 showed occupancy of Fra-2 on the HERV-H LTR region from either the type I or type II subfamily (HERV-H-MC-16, HERV-H-MP20, HERV-H-L19, and HERV-H-Hu13) (Figure 9). The specificity of the occupancy was demonstrated, as Fra-2 was not detected on the β-Globin promoter. These observations underline the role of Fra-2 in activating HERV transcription in ATLL cells.

## 4. Discussion

Several studies have reported HERV gene expression in inflammatory diseases, like multiple sclerosis and rheumatoid arthritis [36,37,38,39], and in cancer, such as breast, prostate, or colon [9,10]. A total of 5% of the HTLV-1-infected patients can develop an inflammatory disease called the HTLV-1-associated myelopathy or paraparesis spastic tropical (HAM/PST) [40,41]. In their studies, Perzova and colleagues show that the sera of HAM/PST patients have high titers of antibodies against HERV-K10 peptides [39,42]. However, in another study, Jones et al. indicate that anti-HERV-K gag and Env-specific T-cell responses were not detected in HAM patients [43], thus suggesting that broad anti-HERV-K immune responses may not be involved in the pathogenesis of HAM. Further studies are still needed to understand if cross-reactive immunity to HTLV-transactivated HERV-K expression in the central nervous system could play a role in the pathogenesis of HAM. However, little is known about the role of HERV in ATLL [31]. In this study, we found that patients with acute ATLL express high levels of antibodies against HERV Env. We also demonstrated that the transcription levels of HERV-H and HERV-K (HML-6) Env genes in ATLL-derived cells and Peripheral Blood Mononuclear Cells (PBMCs) from acute ATLL patients were significantly higher than those in HTLV-1 asymptomatic carriers, showing that the occurrence of ATLL can be associated with the expression of these HERV genes, further supporting that antibodies against HERV Env proteins may be potential biomarkers for the earliest diagnosis of ATLL.

To further confirm the potential role of antibodies against HERV for the diagnosis of the onset of ATLL, we next investigated the presence of humoral response against envelope proteins of several HERVs in the sera of HTLV-1 asymptomatic carriers (ACs) vs. acute ATLL patients. We found that all patients with acute ATLL displayed higher antibody levels against HERV Env compared to HTLV-1 asymptomatic carriers (ACs) or non-infected controls (NI) (Figure 1). To further support the significance of HERV Env-specific humoral response as a possible disease marker for ATLL, it will be interesting to first stratify the population of ATLL patients according to their clinical subtypes since there is no formal grading system to predict ATLL prognosis [2] as a future aim. We will perform complete anti-HERV antibody profiling using Luciferase Immunoprecipitation Systems (LIPS) [44]. Unlike ELISAs, which often screen for responses to peptides derived from diverse HERV families, LIPS is a sensitive method with an extensive dynamic antibody titer range that allows the profiling of antibody responses from patients to antigens associated with infection [44].

We also investigated the molecular mechanism inducing increased HERV expression in ATLL. Cell transformation is often the consequence of mutations in the genome and the accumulation of changes in the epigenome. Tumors can arise from the hierarchical or a cancer stem cell model (CSC) or non-CSCs that acquire progressively oncogenic changes [45,46]. In both models, epigenetic alterations are critically involved [45,46]. Epigenetic alteration is one possible explanation of how HTLV-1 infection may enhance HERV transcription in ATLL cells. If we consider epigenetic alteration that confers plasticity, the member of the polycomb family EZH2 is particularly interesting. EZH2 regulates many genes, including HERV genes, by activating H3K27 trimethylation (H3K27me3) [47]. Interestingly, an aberrant activation of polycomb repressive complex 2 (PRC2), mainly based on an overexpression of EZH2, has been shown in ATLL cells [48,49]. Most downregulated genes associated with the accumulation of H3K27me3 in ATLL were detected at the early stage of disease in either HTLV-1-infected T cells from asymptomatic HTLV-1 carriers or patients with indolent ATLL. Tax modulates the EZH2 function, and a similar H3K27me3 pattern was observed in Tax-immortalized cells [48,49]. However, the inhibition of EZH2 by inhibitors can activate HERV expression in various cancers [47], and Toyoda et al. recently showed that HBZ binds to EZH2 and reduces its activity [50]. These observations suggest that the inhibition of EZH2 by HBZ could be one of the mechanisms that induce HERV transcription in ATLL. However, we could not confirm this hypothesis in our experimental model. Indeed, we did not observe an increase in HERV Env gene expression in cells that stably expressed HBZ (Appendix A). Since the cellular function of EZH2 is context dependent, further investigations are needed to appropriately assess the link between epigenetic modulation and EZH2 function in ATLL.

Current data show that Tax can activate the expression of HERV-K-10, HERV-W8, and HERV-H (MC16) [31]. While 60% of fresh ATLL cells lack tax expression and all ATLL specimens express the hbz gene, it is unclear how HERV LTRs are regulated during the chronic and late stages of the disease. The results presented here provide new insights into understanding the regulation of HERV gene expression in ATLL. Previously, Toufaily et al. proposed that in T cells, the expression of HERV-H and HERV-W genes were activated through many transcription factors, such as NF-kB, NFAT, STAT, AP-1, and CREB [31]. Interestingly, ATLL cells have a specific profile of AP-1 transcription factors presenting a more critical expression of JunD and Fra-2 (also called FosL2) [15].

Using several complementary approaches, such as a luciferase assay and qRT-PCR, we observed that the expression of Fra-2 activates the transcription of HERV-H Env and HERV-K Env. Using a chromatin immunoprecipitation assay, we demonstrate that in an ATLL-derived cell line (ATL-2), Fra-2 binds to the promoter of several HERV-H LTRs, thereby driving their activation. Interestingly, our observations are in agreement with previous studies describing an increase in the abundance of HERV expression in cancer cells that have a heightened expression level of Fra-2, such as prostate cancer (PCa) or colorectal cancer (CRC) [25,26,27,51,52]. Indeed, Kajane et al. have reported that the expression of JunD, Fra-1, and Fra-2 has been associated with a more aggressive clinical outcome in prostate cancer [53]. Furthermore, several studies have reported the presence of serum antibodies against HERV-K gag, an increased expression of HERV-K Env in the PBMCs of PCa patients, and increased levels of HERV-K gag expression in malignant prostate regions in males with PCa [24,26,27,54]. Furthermore, Fra-2 expression is also upregulated in colon carcinomas [55,56,57]. Pérot et al. reported a correlation between HERV-H reactivation and clinical features, such as tumor cells in lymph nodes [58].

## 5. Conclusions

Our findings in this study support the hypothesis of the potential use of HERV-K Env and HERV-H Env antibodies as biomarkers of ATLL disease progression; it may be helpful to more thoroughly evaluate the diagnostic and prognostic values of HERV antibodies. Several studies have used different approaches to investigate the role played by HERVs in cancer pathogenesis, and further analyses are required to understand better how Fra-2 is regulated in ATLL cells and if activation of the different HERVs plays a role in HTLV-1-driven oncogenesis.

## Figures and Tables

**Figure 1 cells-13-01517-f001:**
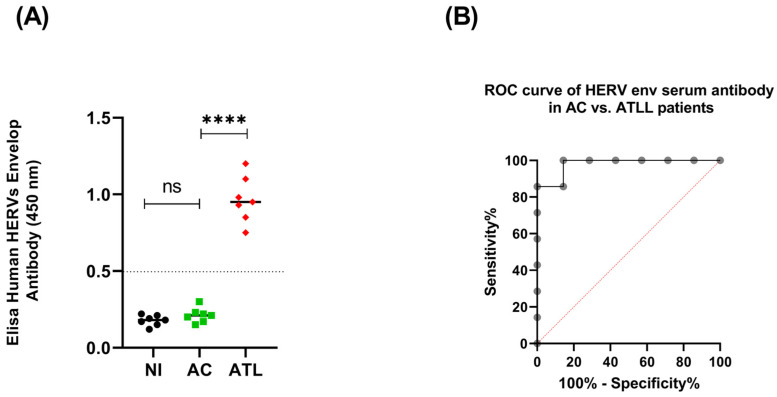
Prevalence of Abs against human HERV Env antigens in patients with acute ATL and the respective control groups. (**A**) HERV envelope (ENV) antigenemia in sera from non-infected patients NI (*n* = 7; black circle), HTLV-1 asymptomatic carriers (*n* = 7; green square), and acute ATLL patients ATLL (*n* = 7; red diamond). The dotted lines represent positivity thresholds calculated by ROC analysis. (**B**) The area under the curve (AUC) and its statistical significance are reported (ns *p* ≤ 0.05 and **** *p* < 0.0001).

**Figure 2 cells-13-01517-f002:**
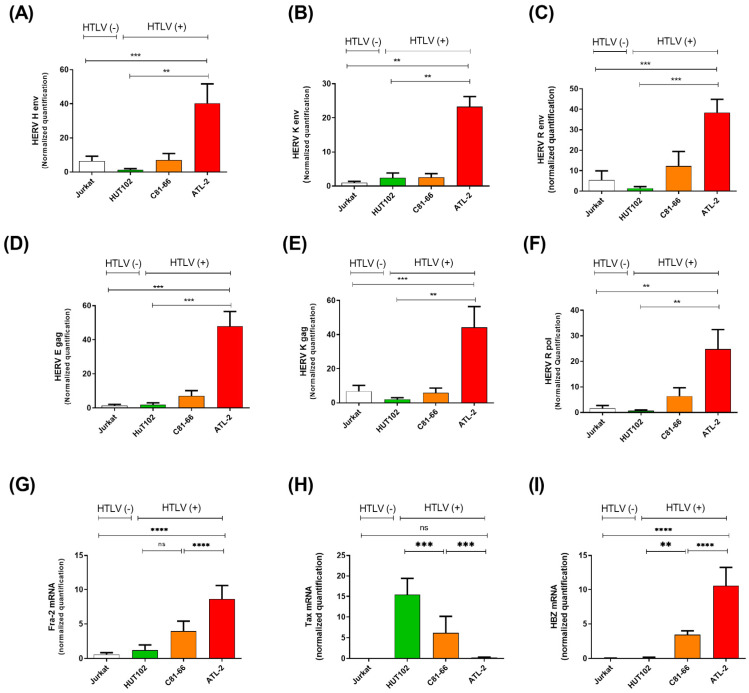
Detection of human HERV mRNA by qRT-PCR in ATL-derived cell lines. (**A**–**C**) Expression of HERV mRNA was assessed in one HTLV-1-negative cell line (Jurkat) and three HTLV-1-derived cell lines (HUT102, C81–66, and ATL-2). (**D**,**E**) HERV gag mRNA expression was compared in Jurkat cells and HTLV-1-derived cell lines. (**F**) HERV-R-Pol mRNA was compared in Jurkat cells and HTLV-1-derived cell lines (** *p* ≤ 0.001). (**G**–**I**) Relative Fra-2, Tax, and HBZ expression in Jurkat cells and HTLV-1-derived cell lines (statistical significance was determined using a one-way ANOVA test with Dunn‘s multiple comparisons post-test ns *p* ≤ 0.05, ** *p* ≤ 0.01; *** *p* ≤ 0.001, **** *p* ≤ 0.0001).

**Figure 3 cells-13-01517-f003:**
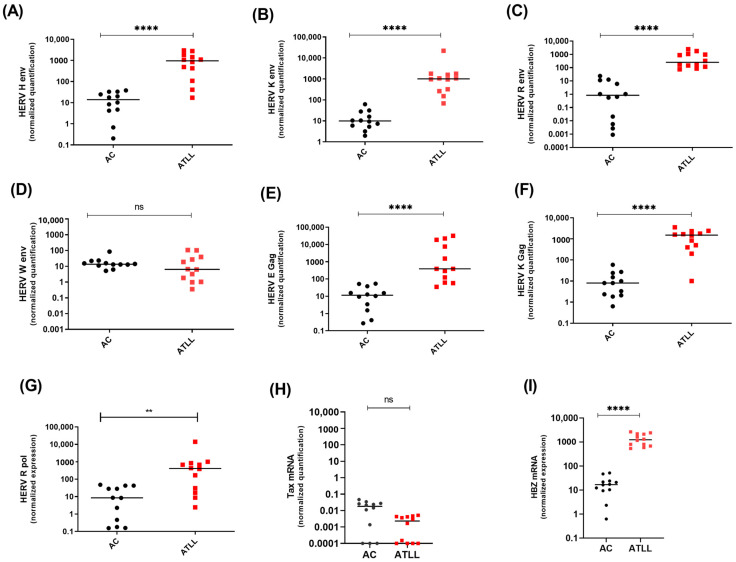
Detection of human HERV mRNA in CD8^+^-depleted PBMCs from HTLV-1-infected patients. (**A**–**D**) HERV mRNA was expressed in acute ATLL patients (red square) compared to HTLV-1 asymptomatic carriers (ACs) (black dot). (**E**,**F**) HERV gag mRNA expression was compared in AC (black dot) and ATLL patients (red square) (one-way ANOVA test with Dunn‘s multiple comparisons post-test ns *p* ≤ 0.05, ** *p* ≤ 0.001; **** *p* ≤ 0.00001). (**G**) HERV-R-Pol was measured in AC and ATLL patients compared to ACs (** *p* ≤ 0.001). (**H**,**I**) The relative expression of Tax and HBZ in AC patients and ATL patients.

**Figure 4 cells-13-01517-f004:**
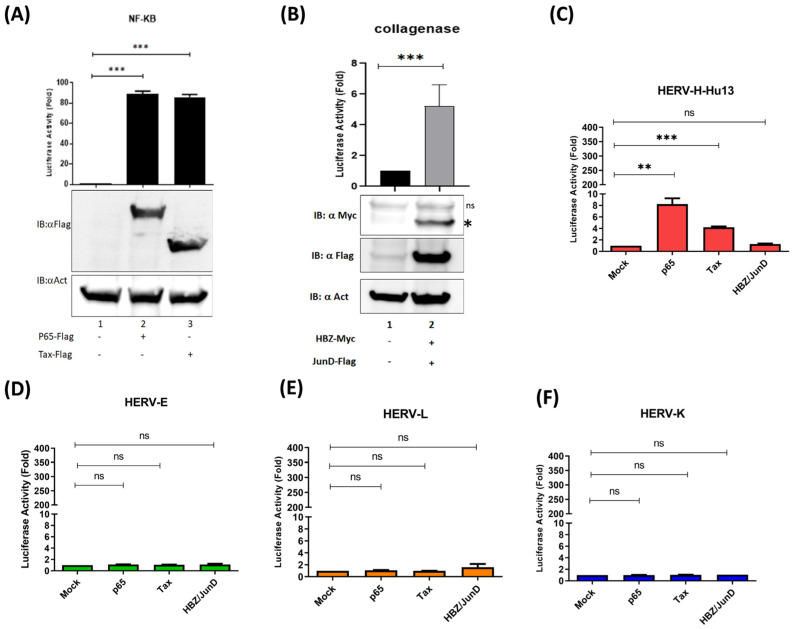
HBZ does not activate HERV LTR. HEK293T was co-transfected with a plasmid carrying the luciferase reporter gene under the control of an NF-kB-dependent promoter (**A**), the collagenase promoter (**B**), or different HERV LTRs (**C**–**F**) and Tax-Flag, p65-Flag, or HBZ-Myc expression vectors, in addition to pRcActin-LacZ for the normalization of transfection efficiency. Cells were harvested 48 h. post-transfection and assayed for luciferase activity. The results show a fold increase compared to the mock control (set at a value of 1) and represent the mean values of three independently transfected cells. (**A**,**B**) Western blot analyses assessed the expression of Tax, p65, and HBZs. Actin is shown as a loading control (one-way ANOVA test with Dunn‘s multiple comparisons post-test ns *p* ≤ 0.05, * *p* ≤ 0.01, ** *p* ≤ 0.001; *** *p* ≤ 0.00001).

**Figure 5 cells-13-01517-f005:**
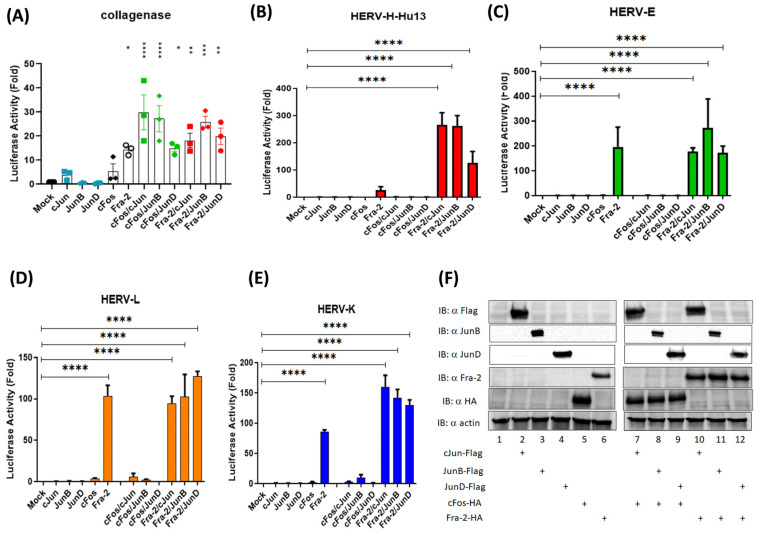
Fra-2 but not cFos activate the HERV LTR. HEK293T cells were co-transfected with a plasmid carrying the luciferase reporter gene under the control of the collagenase promoter in triplicate as shown (**A**) or different HERV 5′LTRs (**B**–**E**), different combinations of binding partners of AP-1-related transcription factors, and pRcActin-LacZ. The cells were harvested 48 h post-transfection and assayed for luciferase activity. The results show a fold increase in the mock control and represent the mean values of three independently transfected cell samples. (**F**) Western blot analyses were carried out to assess the expression of AP-1 transcription factors. Actin is shown as a loading control (one-way ANOVA test with Dunn‘s multiple comparisons post-test ns *p* ≤ 0.05,* *p* ≤ 0.01, ** *p* ≤ 0.001; *** *p* ≤ 0.0001 and **** *p* ≤ 0.00001).

**Figure 6 cells-13-01517-f006:**
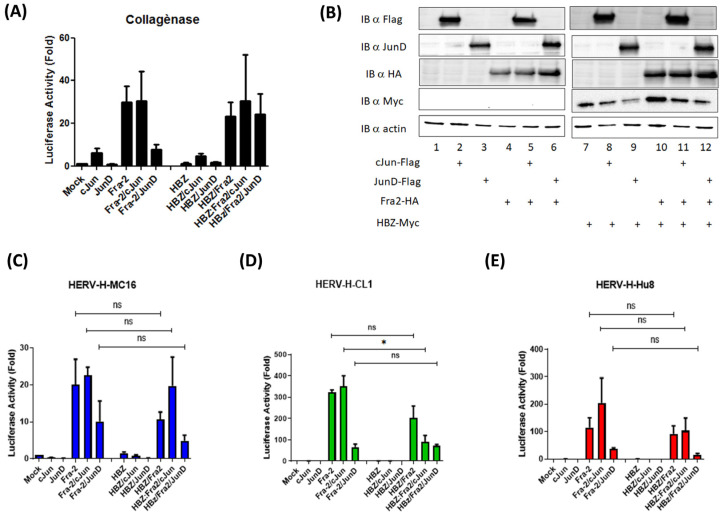
HBZ does not alter the activation of type I and type II HERV-H LTRs by Fra-2. (**A**) HEK293T cells were co-transfected with the different AP-1 in the absence or presence of HBZ and a plasmid carrying the luciferase reporter gene under the control of the collagenase promoter. (**B**) Western blot analyses were carried out to assess the expression of AP-1 transcription factors and HBZ. Actin is shown as a loading control. HEK293T cells were co-transfected with a plasmid carrying the luciferase reporter gene under the control of the collagenase promoter (**A**) or different HERV 5′LTRs (**C**–**E**), different AP-1 expression vectors, and pRcActin-LacZ in the presence or absence of HBZ. The cells were harvested 48 h post-transfection and assayed for luciferase activity. The results show a fold increase in the mock control and represent the mean values of three independently transfected cell samples (one-way ANOVA test with Dunn‘s multiple comparisons post-test ns *p* ≤ 0.05, * *p* ≤ 0.01).

**Figure 7 cells-13-01517-f007:**
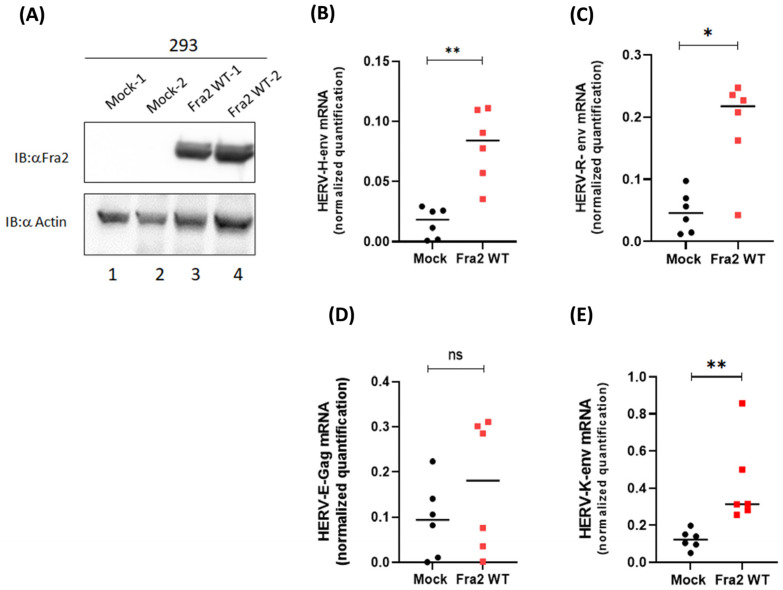
HERV Env mRNA detected in HEK293T cells stably expressing Fra-2. (**A**) Western blot analyses were carried out on the lysate of two pools of HEK293-Fra2 to assess the expression of Fra-2. Beta actin is shown as a loading control. (**B**–**E**) HEK293T stably expressing Fra2 was harvested at different passages (from p3 to p8), and the expression of HERV-H Env, HERV-R Env, HERV-K Env, and HERV-E gag mRNAs was assessed by qRT-PCR (one-way ANOVA test with Dunn‘s multiple comparisons post-test ns *p* ≤ 0.05, * *p* ≤ 0.01; ** *p* ≤ 0.001).

**Figure 8 cells-13-01517-f008:**
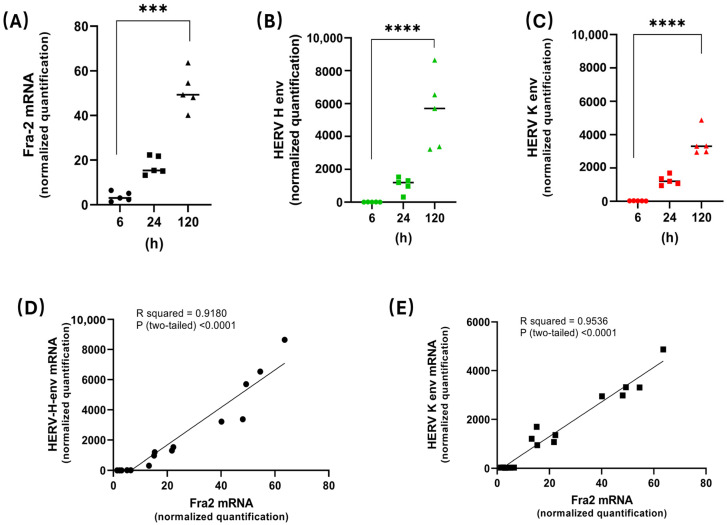
Kinetic analysis of Fra-2 and HERV Env mRNA in CD8^+^-depleted PBMCs from asymptomatic carriers and ATL patients. CD8^+^-depleted PBMCs from five ATL patients were cultivated ex vivo for five days, and Fra-2 (closed black square) (**A**), HERV-H Env (closed green triangle) (**B**), and HERV-K Env mRNA (closed red triangle) (**C**) were quantified at different time points using qRT-PCR (one-way ANOVA test with Dunn‘s multiple comparisons post-test; *** *p* ≤ 0.0001 and **** *p* ≤ 0.00001 (**D**,**E**) Relevance of Fra-2 and HERV Env mRNA expression in ATL patients, as analyzed with the Pearson correlation Test.

**Figure 9 cells-13-01517-f009:**
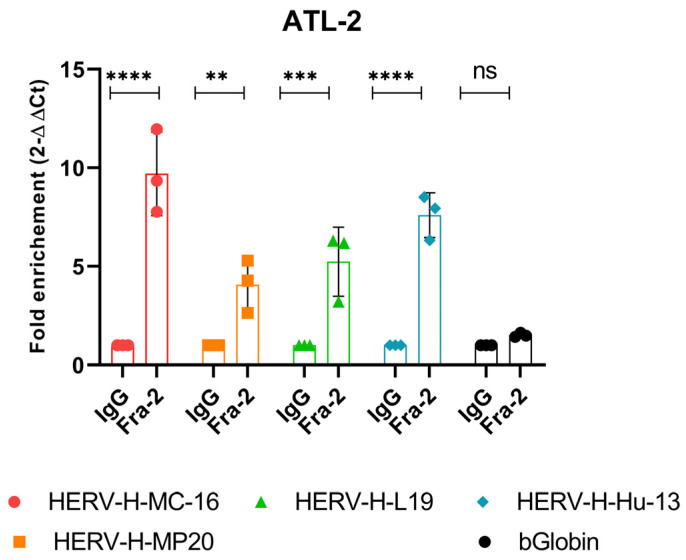
Fra-2 binds to HERV-H-LTR in ATL-2 cells. ChIP assays were performed on chromatin prepared from the indicated ATL-2 cell lines using antibodies against Fra-2. Data are presented as fold enrichment relative to the IgG control. Data are an average of three independent experiments. Error bars represent the SEM (two-way ANOVA *t*-test, ns *p* ≤ 0.05 ** *p* < 0.01, *** *p* < 0.001, **** *p* < 0.0001).

## Data Availability

All data supporting the findings of this study are available in the paper and its Appendix A section. Primer sequences are provided in Appendix A.

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
