# Peer review of "The Oncoprotein Fra-2 Drives the Activation of Human Endogenous Retrovirus Env Expression in Adult T-Cell Leukemia/Lymphoma (ATLL) Patients"

_cells, 2024, doi:10.3390/cells13181517_

Round 1

Reviewer 1 Report

Comments and Suggestions for Authors

The authors propose an article which reports the results of a study providing additional information on the humoral response against HERVs env in sera from ATL patients, compared with asymptomatic carriers living with HTLV-1 and non-infected individuals, and on the expression of HERV genes in HTLV-1 infected cells from ATL cell lines or patients with acute ATL. Results show an upregulation both of the humoral response to HERVs env in samples from ATL patients with respect to asymptomatic carriers living with HTLV-1 and non-infected individuals and of the HERVs gene expression in ATL-derived cell lines and in PBMC from ATL patients in comparison with those from asymptomatic carriers living with HTLV-1.  Moreover, the oncoprotein Fra-2 was found to activate the transcription of several HERV families, including HERV-H and HERV-K families, showing that this protein could be implicated in mechanisms controlling the interaction between HTLV-1 and HERVs in ATL cells. Authors conclude that novel findings of their study could be useful for basic or clinical research on ATL. The proposed article shows some novelties of interest for research and management of HTLV-1 driven diseases. However, in the opinion of these reviewer, this manuscript suffer, in some parts, from a certain overestimation of the obtained results that can be counterproductive to draw attention of the reader to the real, focal points of the study.   Some changes are necessary to ameliorate the manuscript, to avoid confusion in the reader, and to better highlight the limits of the study.

Here are the suggested changes.

Major points

1) Title. The sentence “a key player in the molecular mechanisms of Adult T-cell leukemia/Lymphoma (ATLL)” is improper and misleading. The sentence must be removed from the title to better address the issue on which this study is focused on.

2) Abstract, lines 22-25. The sentence from line 22 “Our study..”, to line 25 “..chemoresistance.”, has absolutely no sense in the context of this study and of the obtained results (“Our study…reveals that HERVs..have provided evolutionary advantages to the host??????)! The entire sentence must be cancelled.

3) Introduction, lines 68-69. This sentence gives a distorted picture of the contents of this study. Remove it. The following sentences are much more informative and adherent to the contents of this study.

4) Introduction, lines 76-77. The part of the sentence from “As increased..” to “…biomarker,” is speculative and useless. Remove it and accordingly modify the next part of the sentence.

5) Introduction, lines 79-80. In this form the last sentence of the “Introduction” seems an inappropriate overestimation, at the moment, of the results of this study. Change it starting with a sentence like “ Further studies will let us know whether anti-HERV antibodies could be…….”

6) Materials and Methods, lines 121-122. It is well known that when comparing gene expression of cells of different malignant/dysfunctional states, such as occurred in this study, by rt-qPCR, the reference housekeeping gene is very important. To avoid results biased by the reference gene, one possibility is comparing the results obtained with more than one housekeeping. Another possibility is that the experimental phase can be preceded by preliminary tests to select an internal control gene which appears to be the most stable within the specific cell types of interest. Authors must choose between the two possibility, produce additional data if not yet available, and modify the manuscript accordingly.  

7) Results, lines 173-174. “could represent a useful diagnostic or prognostic 173 cancer biomarker [19–21]”.  Better “could represent a potential diagnostic or prognostic 173 cancer biomarker [19–21]”. Modify.

8) Results, lines 174-178. The experimental group NI of Figure 1A is never mentioned. Why? This omission must be corrected in the revised version.

9) Results, line 180. Here the terms “controls, Control sera” are utilized for indicating the AC experimental group. This can be misleading (control = NI?) and must be modified.

10) Results, line 184. “a candidate biomarker for developing the disease”. Based on here mentioned results, possibly a diagnostic, not a prognostic biomarker. In this case better  “a candidate biomarker for having developed the disease”.

11) Discussion, lines 355-363. All this introductory part of the “Discussion” seems pleonastic. It diverts the attention of the reader towards aspects of ATLL loosely connected with this study. It must be simply cancelled.

12) Discussion, line 365. Reference 41 (Perzova R. et al. 2015. Increased seroreactivity….) is not properly mentioned. In fact, this article refers to anti-HERVs protein sera levels in HTLV-1 patients and not to HERV gene expression. Thus, even if not properly mentioned for HERV gene expression, the study of Perzova et al.  greatly deals with results reported in this manuscript concerning immunoreactivity of sera from ATL patients towards HERVs genes. Notably, the same group published data on the same subject two years before (Perzova R., et al. Increased seroreactivity to HERV-K10 peptides in patients with HTLV myelopathy. Virol J. 2013 Dec 23;10:360), thus anticipating by about 10 years results reported in the present study! Interestingly, however, results reported in the same year by another group are, apparently, not in accordance with results of Perzova et al. and with those of the present study (R Brad Jones, et al. Human Endogenous Retrovirus K(HML-2) Gag and Env specific T-cell responses are not detected in HTLV-I-infected subjects using standard peptide screening methods. Journal of Negative Results in BioMedicine 2013, 12:3).

All of the above cannot be ignored in the “Discussion” of this manuscript. The above mentioned published articles must be inserted in the reference list and the section “Discussion” must be accordingly revised.

13) Discussion, generalThe novelty of the putative role of Fra-2 in activation of HERVs expression in ATL patients is rightly highlighted. However, the major limit of the study, consisting of the total ignorance of how Fra-2 is modulated in HTLV-1 infected malignant cells (see negative results for HBZ), is not mentioned. Is it a “side-effect” related to a  nonspecific transcriptional activation in leukemic cells leading to a similarly “side-effect” activation of HERVs expression that has nothing to do with the HTLV-1-driven oncogenesis? A mention of this major limit and to the need for future research on this aspect must be made in the Discussion.  

Minor

1) line 52. This is the first mention of HERVs: add the not shortened term.

2) line 179. “We next use…”. Correct  “We next used…”.

3) line 339.  “We next assess…”. Correct “We next assessed…”.

Author Response

The authors propose an article which reports the results of a study providing additional information on the humoral response against HERVs env in sera from ATL patients, compared with asymptomatic carriers living with HTLV-1 and non-infected individuals, and on the expression of HERV genes in HTLV-1 infected cells from ATL cell lines or patients with acute ATL. Results show an upregulation both of the humoral response to HERVs env in samples from ATL patients with respect to asymptomatic carriers living with HTLV-1 and non-infected individuals and of the HERVs gene expression in ATL-derived cell lines and in PBMC from ATL patients in comparison with those from asymptomatic carriers living with HTLV-1.  Moreover, the oncoprotein Fra-2 was found to activate the transcription of several HERV families, including HERV-H and HERV-K families, showing that this protein could be implicated in mechanisms controlling the interaction between HTLV-1 and HERVs in ATL cells. Authors conclude that novel findings of their study could be useful for basic or clinical research on ATL. The proposed article shows some novelties of interest for research and management of HTLV-1 driven diseases. However, in the opinion of these reviewer, this manuscript suffer, in some parts, from a certain overestimation of the obtained results that can be counterproductive to draw attention of the reader to the real, focal points of the study.   Some changes are necessary to ameliorate the manuscript, to avoid confusion in the reader, and to better highlight the limits of the study.

Here are the suggested changes.

We thank the reviewer for his/her encouraging comments

Major points

1) Title. The sentence “a key player in the molecular mechanisms of Adult T-cell leukemia/Lymphoma (ATLL)” is improper and misleading. The sentence must be removed from the title to better address the issue on which this study is focused on.

Our Response:  This has been corrected.

2) Abstract, lines 22-25. The sentence from line 22 “Our study..”, to line 25 “..chemoresistance.”, has absolutely no sense in the context of this study and of the obtained results (“Our study…reveals that HERVs..have provided evolutionary advantages to the host??????)! The entire sentence must be cancelled.

Our Response:  This has been corrected.

3) Introduction, lines 68-69. This sentence gives a distorted picture of the contents of this study. Remove it. The following sentences are much more informative and adherent to the contents of this study.

Our Response:  This has been corrected.

4) Introduction, lines 76-77. The part of the sentence from “As increased..” to “…biomarker,” is speculative and useless. Remove it and accordingly modify the next part of the sentence.

Our Response:  This has been corrected.

5) Introduction, lines 79-80. In this form the last sentence of the “Introduction” seems an inappropriate overestimation, at the moment, of the results of this study. Change it starting with a sentence like “ Further studies will let us know whether anti-HERV antibodies could be…….”

Our Response:  This has been corrected.

6) Materials and Methods, lines 121-122. It is well known that when comparing gene expression of cells of different malignant/dysfunctional states, such as occurred in this study, by rt-qPCR, the reference housekeeping gene is very important. To avoid results biased by the reference gene, one possibility is comparing the results obtained with more than one housekeeping. Another possibility is that the experimental phase can be preceded by preliminary tests to select an internal control gene which appears to be the most stable within the specific cell types of interest. Authors must choose between the two possibility, produce additional data if not yet available, and modify the manuscript accordingly.  

Our Response:  This has been corrected. We normalized our results with two housekeeping genes (HPRT-1 and GADPH); no significant differences were observed, and the manuscript has been modified accordingly

7) Results, lines 173-174. “could represent a useful diagnostic or prognostic 173 cancer biomarker [19–21]”.  Better “could represent a potential diagnostic or prognostic 173 cancer biomarker [19–21]”. Modify.

Our Response:  This has been corrected.

8) Results, lines 174-178. The experimental group NI of Figure 1A is never mentioned. Why? This omission must be corrected in the revised version.

Our Response:  This has been corrected.

9) Results, line 180. Here the terms “controls, Control sera” are utilized for indicating the AC experimental group. This can be misleading (control = NI?) and must be modified.

Our Response:  This has been corrected.

10) Results, line 184. “a candidate biomarker for developing the disease”. Based on here mentioned results, possibly a diagnostic, not a prognostic biomarker. In this case better  “a candidate biomarker for having developed the disease”.

Our Response:  This has been corrected.

11) Discussion, lines 355-363. All this introductory part of the “Discussion” seems pleonastic. It diverts the attention of the reader towards aspects of ATLL loosely connected with this study. It must be simply cancelled.

Our Response:  This has been corrected.

12) Discussion, line 365. Reference 41 (Perzova R. et al. 2015. Increased seroreactivity….) is not properly mentioned. In fact, this article refers to anti-HERVs protein sera levels in HTLV-1 patients and not to HERV gene expression. Thus, even if not properly mentioned for HERV gene expression, the study of Perzova et al.  greatly deals with results reported in this manuscript concerning immunoreactivity of sera from ATL patients towards HERVs genes. Notably, the same group published data on the same subject two years before (Perzova R., et al. Increased seroreactivity to HERV-K10 peptides in patients with HTLV myelopathy. Virol J. 2013 Dec 23;10:360), thus anticipating by about 10 years results reported in the present study! Interestingly, however, results reported in the same year by another group are, apparently, not in accordance with results of Perzova et al. and with those of the present study (R Brad Jones, et al. Human Endogenous Retrovirus K(HML-2) Gag and Env specific T-cell responses are not detected in HTLV-I-infected subjects using standard peptide screening methods. Journal of Negative Results in BioMedicine 2013, 12:3).

All of the above cannot be ignored in the “Discussion” of this manuscript. The above mentioned published articles must be inserted in the reference list and the section “Discussion” must be accordingly revised.

Our Response:  In their studies, Perzova and colleagues, as well as Jones, have investigated the response against HERV-K peptides in HTLV-1 patients who have developed an inflammatory disease called HTLV-associated myelopathy (HAM/PST) and not in patients who have developed an adult T cell leukemia/Lymphoma. The pathological processes of both diseases are very different. Results reported that HAM/PST patients do not allow us to anticipate whether it will also be valid for the ATLL patients. However, we have corrected the discussion and have inserted the mentioned paper in the reference list.

13) Discussion, generalThe novelty of the putative role of Fra-2 in activation of HERVs expression in ATL patients is rightly highlighted. However, the major limit of the study, consisting of the total ignorance of how Fra-2 is modulated in HTLV-1 infected malignant cells (see negative results for HBZ), is not mentioned. Is it a “side-effect” related to a  nonspecific transcriptional activation in leukemic cells leading to a similarly “side-effect” activation of HERVs expression that has nothing to do with the HTLV-1-driven oncogenesis? A mention of this major limit and to the need for future research on this aspect must be made in the Discussion.  

Our Response:  This has been corrected.

Minor

1) line 52. This is the first mention of HERVs: add the not shortened term.

Our Response:  This has been corrected.

2) line 179. “We next use…”. Correct  “We next used…”.

Our Response:  This has been corrected.

3) line 339.  “We next assess…”. Correct “We next assessed…”.

Our Response:  This has been corrected.

Reviewer 2 Report

Comments and Suggestions for Authors

The authors first demonstrated that HERV can serve as a prognostic marker for diagnosing ATLL, providing evidence of high HERV expression in both primary ATLL patient samples and ATL cell lines. However, the viral HBZ and TAX genes do not play a role in HERV LTR transcription. Further investigations revealed that the heterodimers Fra-2/cJun, Fra-2/JunB, and Fra-2/JunD transactivate HERV LTR by directly binding to specific HERV LTR loci.

Overall, the manuscript is interesting and well performed. Conclusion are supported by the data.

Minor comments:

1)      Did the authors study Fra-2 expression in primary ATLL patient samples?

2)      Could the authors provide details on how to use the in-silico prediction model to identify the AP-1 binding sites in the U3 and R regions of the LTR?

3)      The authors should verify the anti-actin antibodies used. In the Methods section (Sigma, A3853), the anti-actin antibody detects total actin. However, in the manuscript, the authors refer to beta-actin in line 315 and to alpha-actin in all the immunoblot figures.

4)      The authors conducted ELISA assays in HEK293T cells. Why did authors not perform ELISA in both HTLV-1 negative (e.g. Jurkat cell) and HTLV-1 positive leukemic cell lines? Is it possible that HERV LTR transcription depends on other transcription factors present in leukemic cells?

5)      The authors performed Fra-2 ChIP in ATL-2 cell line. It would be beneficial to include Fra-2 ChIP data from both normal cells and primary ATLL patient samples, with either HTLV-1 positive or negative.

6)      If Fra-2 is highly expressed in HTLV-1 positive cells, could it influence leukemogenesis in vivo?

The manuscript contains several misspellings that should be corrected. For example:

Line 167: Could the authors confirm whether it is the Mann-Whitney U-test instead of the Mann-Whitney T-test?

Line 209: "HTLV-1-derived cell lines (F). HERV-" should be replaced with "HTLV-1-derived cell lines. (F) HERV-."

Line 254: "HERV LTRs (C-G)" should be corrected to "HERV LTRs (C-F)" as there is no figure G in Figure 4.

Lines 275 and 278: "Supplemental Figure" should be "3A, 3B-G" instead of "2A, 2B-2G," respectively.

Line 291: There is a missing figure legend for Figure 6A.

Line 371: The authors should check whether "HERV genes" is misspelled as "HER gene."

Author Response

The authors first demonstrated that HERV can serve as a prognostic marker for diagnosing ATLL, providing evidence of high HERV expression in both primary ATLL patient samples and ATL cell lines. However, the viral HBZ and TAX genes do not play a role in HERV LTR transcription. Further investigations revealed that the heterodimers Fra-2/cJun, Fra-2/JunB, and Fra-2/JunD transactivate HERV LTR by directly binding to specific HERV LTR loci.

Overall, the manuscript is interesting and well performed. Conclusion are supported by the data.

our response: We thank the reviewer for his/her encouraging comments

Minor comments:

1)      Did the authors study Fra-2 expression in primary ATLL patient samples?

Our Response:  in Terol et al. 2017, we reported that Fra-2 was overexpressed in CD8-depleted PBMCs from ATLL patients. In this manuscript, we have also performed a kinetic and followed of Fra-2 expression in CD8-depleted PBMCs of ATLL patients, and we observed that fra-2 expression is increasing over time in ATLL cells (Fig8A)

2)      Could the authors provide details on how to use the in-silico prediction model to identify the AP-1 binding sites in the U3 and R regions of the LTR?

Our Response:  To elucidate the mechanism by which transcription of the different HERVs was upregulated in ATLL cells, we examined the potential binding of transcription factors throughout the different HERV-LTR regions using Tfsitescan and PROMO. This has been corrected in the manuscript.

3)      The authors should verify the anti-actin antibodies used. In the Methods section (Sigma, A3853), the anti-actin antibody detects total actin. However, in the manuscript, the authors refer to beta-actin in line 315 and to alpha-actin in all the immunoblot figures.

Our Response: Indeed, we have detected the total actin with the antibody A3853 (Sigma); this has been corrected.

4)      The authors conducted ELISA assays in HEK293T cells. Why did authors not perform ELISA in both HTLV-1 negative (e.g. Jurkat cell) and HTLV-1 positive leukemic cell lines? Is it possible that HERV LTR transcription depends on other transcription factors present in leukemic cells?

Our Response: The Elisa test was only performed on Sera derived from non-infected patients (NI), asymptomatic carriers (AC), and ATLL patients to determine the presence of antibodies against HERV env proteins. We believe that the reviewer wants to mention the luciferase assay test. We wanted to perform on T cell lines. Unfortunately, all the T-cell lines used in this study express high levels of fra-2 compared to primary CD+ T cells, but we observed that similarly to primary ATL CD4 T cells, the ATL-2 cell lines also express the AP-1 factor JunD. Our attempt to decrease Fra-2 expression using siRNA strongly decreases the cell viability. Since our analysis of TF binding sites in the HERV-LTR showed sites for Oct-1, Myc, Sox9, or HNF-4, we cannot rule out that other transcription factors could also be involved in regulating HERV-LTR.

5)      The authors performed Fra-2 ChIP in ATL-2 cell line. It would be beneficial to include Fra-2 ChIP data from both normal cells and primary ATLL patient samples, with either HTLV-1 positive or negative.

Our Response: Including ChIP data from primary ATL patient samples would have been interesting; however, performing ChIP from Primary ATLL patients would have required a significant number of ATLL cells per experiment, which we did not have to perform this experiment. Furthermore, fresh or cryopreserved cells from ATLL patients are challenging to obtain. Thus, we choose to perform the ChIP experiment using a patient-directly derived ATL-2 cell line.

6)      If Fra-2 is highly expressed in HTLV-1 positive cells, could it influence leukemogenesis in vivo?

Our Response: Indeed, we believe that the increase in the expression of Fra-2 is influencing the leukemogenic. We are currently preparing another manuscript on the role of increased expression of Fra-2 in ATLL pathogenesis.

The manuscript contains several misspellings that should be corrected. For example:

Line 167: Could the authors confirm whether it is the Mann-Whitney U-test instead of the Mann-Whitney T-test?

Our Response:  We have performed a Mann-Whitney T-test. This has been corrected.

Line 209: "HTLV-1-derived cell lines (F). HERV-" should be replaced with "HTLV-1-derived cell lines. (F) HERV-."

Our Response:  This has been corrected.

Line 254: "HERV LTRs (C-G)" should be corrected to "HERV LTRs (C-F)" as there is no figure G in Figure 4.

Our Response:  This has been corrected.

Lines 275 and 278: "Supplemental Figure" should be "3A, 3B-G" instead of "2A, 2B-2G," respectively.

Our Response:  This has been corrected.

Line 291: There is a missing figure legend for Figure 6A.

Our Response:  This has been corrected.

Line 371: The authors should check whether "HERV genes" is misspelled as "HER gene."

Our Response:  This has been corrected.

Round 2

Reviewer 1 Report

Comments and Suggestions for Authors

All the points raised have been properly addressed and the manuscript is now suitable for publication